# Multifractal Correlation between Terrain and River Network Structure in the Yellow River Basin, China

**Zilong Qin [1] and Jinxin Wang [2,*]**

[1]  School of Remote Sensing and Information Engineering, Wuhan University, Wuhan 430079, China
[2]  School of Earth Science and Technology, Zhengzhou University, Zhengzhou 450001, China
*  Correspondence: jxwang@zzu.edu.cn; Tel.: +86-135-2687-1639

**Abstract:** As the most basic physical geographic elements, basin terrain and river networks have high spatial complexity and are closely related. However, there is little research on the correlation between terrain and river networks. In this paper, the Yellow River Basin was selected as the study area. Topographic factors of multiple dimensions were calculated. The influence of different topographic factors on the river network structure at different scales and their correlation from a multifractal perspective based on geographical detectors and a geographically weighted regression model were determined. The explanatory power of topography on the river network structure at different scales was: multifractal spectrum width > multifractal spectrum difference > slope > average elevation > elevation maximum > elevation minimum, which generally indicated that the topographic factor that has the greatest influence on the river network structure is the complexity and singularity of the terrain. The second-order clustering of regression coefficients from the results of the geographically weighted regression model revealed that the Yellow River basin was divided into three types of high-aggregation areas, which are dominated by the Qinghai-Tibet Plateau, the Loess Plateau, and the Huang-Huaihai Plain, respectively. The clustering results also revealed that the river network structure was affected by different key topographic factors in the different types of areas. This research studies and quantifies the relationship between basin topography and river network structure from a new perspective and provides a theoretical basis for unraveling the development of topography and river networks.

**Keywords:** multifractal analysis; terrain; river networks; quantitative description; correlation analysis; Yellow River basin

## 1. Introduction

Topography is one of the most important elements of the Earth's surface system [1], influencing the characteristics of other natural elements and having a direct impact on human activities. This is why it is one of the fundamental components of geography research [2]. Similarly, as one of the core components of basins, river networks are constantly developing under various natural factors and human activity [3]. An accurate quantitative representation of river network structural features is essential for studying sedimentation processes [4], extreme hydrological events in the basin [5], and river network development [6]. However, accurately quantifying the structural characteristics of the river network and watershed terrain is a key challenge in current geoscience research [7]. Therefore, it is important to quantify the characteristics of watershed terrain and river networks and explore their relationship to study the development and evolution processes of the Earth's surface.

A single-factor or multifactor combination is mainly used to quantify watershed terrain and river network features. The extraction and analysis of geomorphic features based on digital elevation model (DEM) data are currently the most common methods. The DEM is a digital representation of terrain that contains the most essential topographic and

geomorphological information [8], and the watershed river network is often extracted from the DEM. The main topographic indicators in watershed geomorphological analysis include mean elevation, slope [9,10], slope aspect [10], slope length [11], topographic relief [12], topographic roughness [13], and the area elevation integral [14]. For the description of river network morphological characteristics, commonly used indicators include river network density [15–17], branching ratio [18], lateral branching ratio [19], total length, curvature, and backbone area length ratio [20].

Although a single quantitative indicator can reflect the characteristics of watershed topography or water systems from a specific perspective, it is not sufficiently comprehensive to describe their features as a whole [21]. Moreover, because of the complex and variable spatial distribution of watershed topography and river systems with strong singularity, conventional quantitative parameters have certain limitations in describing and studying their characteristics [22,23]. The fractal theory provides the possibility for quantitative characterization of topography and river network features. It is an important branch of the nonlinear discipline. Mandelbrot first proposed a description of the irregularity and self-similarity of entities in nature by measuring the length of the British coastline [24]. The spatial distribution of natural elements (river networks, mountains, topography, and coastlines) is complex and variable. On the other hand, the fractal theory can reveal that they still have mathematical patterns that can be described quantitatively in terms of fractal dimensions [25,26]. Multifractal theory [27] is used to describe many complex evolutionary processes and the morphological characteristics of entities in nature [28]. Compared with mono-fractals, multifractal characteristics are calculated based on the probability of feature information with order moments. They reflect the complexity of the morphological characteristics of the research object with more comprehensive details by the continuous function of fractal dimensions.

As the most basic and important natural geographic elements, the structure and morphological characteristics of watershed topography and river networks are highly complex in their spatial distribution. It is usually difficult to describe their characteristic information quantitatively using conventional methods. Multifractal analysis can quantify the structural characteristics of watershed topography and river networks more comprehensively. As important parts of a basin, the topography and water systems are closely related. For example, topographic relief affects the formation and flow direction of rivers, and rivers erode the topography and affect topographic development. However, no research has been conducted specifically on the mechanism of the interaction between topography and river networks. The Yellow River is the second-longest river in China [29]. The Yellow River basin has great differences in elevation, temperature, precipitation, topography, and water system characteristics in each region [30]. Due to the complex structure and morphological characteristics of the Yellow River Basin, it is of great significance to study the correlation and interaction mechanism between its topography and the river network.

Based on the above, we studied the Yellow River Basin using the multifractal method as the theoretical basis and combining topography, GIS, hydrology, and other disciplines. We aimed to analyze the spatial distribution of the multifractal features of the basin topography and river network at different scales, to unravel the basin topography and river network characteristics and their correlation, and to further analyze the topographic factors that have a greater impact on the river network based on the results of the multifractal analysis. Our research provides a scientific reference and a new perspective for studying the development of the watershed, which has important theoretical significance.

## 2. Materials and Methods

### 2.1. Study Area

The Yellow River Basin is located between 32° N, 96° E and 42° N, 119° E, and it contains more than 370 counties in nine provinces of China (Figure 1). The Yellow River is 5464 km long with a basin area of 795,000 km$^2$, making it the fifth-longest river in the world and the second-longest in China [29]. There are many mountains in the

Yellow River Basin, with a great difference in height between the east and west (the maximum elevation difference is over 6000 m). The western region is mainly composed of a series of high mountains, with an average elevation of more than 4000 m, and perennial snow and glacier landforms; the central region is a loess landform, with serious soil erosion and elevation between 1000–2000 m; the eastern region is mainly composed of the Yellow River mainstream and a small number of low mountain hills, with relatively low terrain. The Yellow River Basin is located in the middle latitudes and is affected by atmospheric and monsoon circulation. Therefore, the climates of different regions in the Yellow River Basin vary significantly. Precipitation is concentrated, unevenly distributed, and exhibits inter-annual variation, with a maximum and minimum annual precipitation ratio of approximately 1.7 to 7.5. The distribution and structure of the river network in the Yellow River basin are relatively complex, mainly including the tributaries of the Weihe, Fenhe, Taohe, and Luohe Rivers. Its prominent characteristics are "less water and more sediment, different sources of water and sediment." The annual average natural runoff of the entire river is 58 billion cubic meters, which accounts for only 2% of the total fluvial runoff in China.

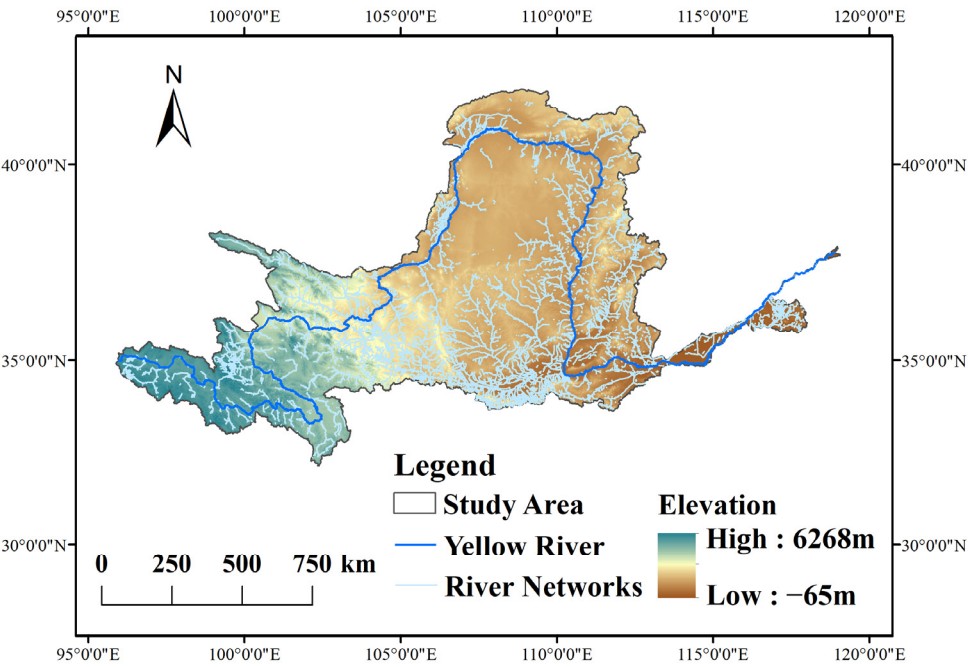

**Figure 1.** The location of the Yellow River Basin.

*2.2. Data Description*

The data used in this study mainly include DEM and river network vector data for the Yellow River Basin. The DEM data were obtained from the NASA EARTH DATA sharing website (https://earthdata.nasa.gov, accessed on 10 July 2022) in the format of ASTER GDEM V3 with a resolution of 30 m. Based on the above DEM data, ArcGIS 10.4 was used to calculate the topographic factors of the study area, including slope, topographic relief, topographic roughness, and maximum and minimum elevation of the area. River network vector data were obtained from the National Center for Basic Geographic Information (https://www.webmap.cn/, accessed on 10 July 2022) and the Earth System Science Data Sharing Platform of the National Earth System Science Data Center (http://www.geodata.cn/, accessed on 10 July 2022). To ensure the accuracy of the river network data, the above river network data were added, deleted, and corrected based on the remote sensing images of Google Earth. In addition, to eliminate the influence of artificial administrative boundaries on the research results, the Yellow River basin was divided into 80 km × 80 km and 40 km × 40 km grids, and the number of statistical units

containing the Yellow River Basin was 262 and 893, respectively. Based on the divided data, the correlation between landforms and the development of the river network was further studied.

*2.3. Method*

2.3.1. Multifractal Analysis

The fractal theory is widely used to describe the characteristics of irregular and complex entities. Monofractals can describe the complexity of a fractal set using a single fractal dimension, which explains the number of spaces filled by fractals without considering local differences in density. The multifractal theory explains the distribution of measurements $[p_i(e)]$ on the fractal set and can divide regions into high- and low-probability regions based on the probability-of-measure distribution [31]. Therefore, multifractals can describe the complexity of a research object in a more detailed and continuous manner [32].

Multifractals are composed of several or more singular monofractal sets, with different fractal behaviors, which are related and have their own fractal dimensions [33,34]. The generalized fractal dimension ($D_q$) and multifractal spectrum [$f(\alpha)$] are mainly used in the description and quantification of object features by multifractals [35]. In this study, fixed-size algorithms (FSA) [36] were used to calculate the multifractal characteristics of the topography and river networks in various statistical units of the Yellow River Basin.

To calculate the multifractal features of the study object, the study area was first covered using boxes of size $e \times e$. $N(e)$ denotes the number of nonempty boxes and is represented by $N(e)$. The probability measure of each unit, that is, the distribution probability of the characteristic information, is denoted by $p_i(e)$, and the calculation formula is shown in Equation (1).

$$p_i(e) = \frac{c_i}{\sum_{i=1}^{N(e)} c_i},\tag{1}$$

where $c_i$ denotes the characteristic information of the study object in box $i$. In this study, $c_i$ denotes the slope of the terrain and the length of the river network. The partition function $M(e,q)$ is then defined as the weighted sum of the slope and river network distribution probability $p_i(e)$ to the power of $q$ (Equation (2)).

$$M(e,q) = \sum_{i=1}^{N(e)} p_i^q(e),\tag{2}$$

where $q$ is the order of the statistical moment and, in general, $q$ ranges from $-\infty$ to $+\infty$. In multifractals, $q$ is used to characterize the degree of inhomogeneity.

The different values of $q$ represent the importance of different probability subsets in the partition function. At a given order moment $q$, the mass exponential function $\tau(q)$ is defined as Equation (3). We calculated the partition function $M(e,q)$ for different box sizes by changing the size of the box under the corresponding $q$ value. Then, $\tau(q)$ can be computed through the coefficient of the straight fitted line of ln $M(e,q)$~ln $e$ [Equation (4)]. Furthermore, we changed the value of $q$ and repeated the above steps to calculate $\tau(q)$ corresponding to different values of $q$.

$$M(e,q) \propto e^{\tau(q)},\tag{3}$$

$$\tau(q) = \lim_{e \to 0} \frac{\ln M(e,q)}{\ln e},\tag{4}$$

Finally, the generalized fractal dimension $D_q$, singularity exponent $\alpha(q)$, and multifractal spectrum $f(\alpha)$ were calculated based on the above results using Equations (5) and (6).

$$D_q = \begin{cases} \frac{1}{q-1} \lim\limits_{e \to 0} \frac{\ln M(e,q)}{\ln e} = \frac{\tau(q)}{q-1} & q \neq 1 \\ \lim\limits_{e \to 0} \frac{\sum_{i=1}^{N(e)} p_i \ln p_i}{\ln e} & q = 1 \end{cases},\tag{5}$$

$D_q$ varies with $q$. When $q = 0$, $D_{q=0}$ denotes the capacity dimension in the fractal dimension; when $q = 1$, $D_{q=1}$ is the information dimension; and when $q = 2$, $D_{q=2}$ is the correlation dimension. In general, $D_q$ is a strictly monotonically decreasing function of $q$. When $q \rightarrow +\infty$, $D_q$ describes the scalar behavior of the region with the most concentrated probability measures in the study area, and when $q \rightarrow -\infty$, $D_q$ describes the scalar behavior of the region with the sparsest probability measures.

$$\begin{cases} \alpha(q) = \frac{d\tau(q)}{dq} \\ f(\alpha) = q \cdot \alpha(q) - \tau(q) \end{cases}, \tag{6}$$

where $\alpha(q)$ represents the singularity of the probability density, and $\alpha(q)$ is a value if the measure of the studied object is evenly distributed. $f(\alpha)$ is typically a smooth up-convex curve, and each point on the $\alpha(q) \sim f(\alpha)$ curve represents the fractal dimension of the same subset of the singularity exponent $\alpha(q)$ [37,38].

Before calculating the multifractal characteristics of the research object, it was necessary to determine whether it had multifractal properties. Generally, the following two indicators are verified to determine whether or not they have the properties of multifractals: (1) the double logarithmic curve (ln $M(e, q)$-ln $e$) of the partition function and scale $e$ has a good linear relationship, which means that the research object has the properties of multifractals; (2) if $\tau(q)$ is a convex function of $q$, then the object of study has the properties of multifractals.

When analyzing the multifractal characteristics of the study object, we must calculate the width of the multifractal spectrum $\Delta\alpha$ and the difference in the multifractal spectrum $\Delta f$ [Equations (7) and (8)]. $\Delta\alpha$ can quantitatively describe the degree of inhomogeneity of the basin topography/river network, and $\Delta f$ can be used to express the difference in quantity between the maximum distribution probability and minimum probability subset of the basin feature information. Its specific meaning can be found in the literature [39].

$$\Delta\alpha = \alpha_{\max} - \alpha_{\min}, \tag{7}$$

$$\Delta f = f(\alpha_{\min}) - f(\alpha_{\max}) \tag{8}$$

### 2.3.2. The Geographical Detectors

Spatial differentiation is a fundamental characteristic of geographical phenomena. A geographical detector detects the spatial differentiation of geographical elements and reveals their driving factors, as proposed by Wang et al. [40]. Detection using geographical detectors [41] includes differentiation and factor detection, interaction detection, risk area detection, and ecological detection. According to the principles and application fields of geographical detectors, we used the first detector to analyze the influence and correlation of topography on river network development.

The differentiation and factor detector is used to detect the spatial differentiation of $Y$ (river network) and the extent to which each factor $X$ (topography) explains the spatial differentiation of $Y$. The degree of interpretation is measured using the $q$ value based on the following equation:

$$q = 1 - \frac{\sum_{h=1}^{L} N_h \sigma_h^2}{N\sigma^2} = 1 - \frac{SSW}{SST}, \tag{9}$$

where $h = 1, 2, \cdots$; $L$ is the stratification of variable $Y$ or factor $X$, that is, classification or partition; and $N_h$ and $N$ are the number of units in layer $h$ and the whole area, respectively. $\sigma_h^2$ and $\sigma^2$ are the variances of $Y$ values in layer $h$ and the entire area, respectively. $SSW$ and $SST$ are the sum of variances within layers and the total variance of the entire region, respectively. The range of $q$ is $[0, 1]$. A larger $q$ value indicates that the spatial heterogeneity of $Y$ (river network) is more evident. If stratification is generated by variable $X$, the larger the $q$ value, the stronger the explanatory power of the independent variable $X$ to attribute $Y$; otherwise, the weaker it is. In the extreme case, when $q$ is 1, it indicates that variable $X$

can completely control the spatial distribution of $Y$; when $q$ is 0, it indicates that variable $X$ has no relationship with $Y$. The $q$ value indicates that the variable $X$ explains $100 \times q\%$ of $Y$.

A transformation of the $q$ values satisfies the non-central $F$ distribution:

$$F = \frac{N-L}{L-1}\frac{q}{1-q} \sim F(L-1, N-L; \lambda), \tag{10}$$

$$\lambda = \frac{1}{\sigma^2}\left[\sum_{h=1}^{L}\overline{Y}_h^2 - \frac{1}{N}\left(\sum_{h=1}^{L}\sqrt{N_h}\overline{Y}_h\right)^2\right], \tag{11}$$

where $\lambda$ is the non-central parameter and $\overline{Y}_h$ is the mean value of layer $h$. We can test whether the $q$ value is significant using Equation (11).

### 2.3.3. Geographically Weighted Regression

Geographically Weighted Regression (GWR) is a spatial regression model based on local smoothness that can effectively estimate data with spatial autocorrelation and reflect the spatial heterogeneity of variables in different regions [42,43]. The equation for the GWR model is as follows:

$$y_i = \beta_0(u_i, v_i) + \sum_{k=1}^{p}\beta_k(u_i, v_i)x_{ik} + \varepsilon_i, \tag{12}$$

where $y_i$ is the dependent variable; $\beta_0$ is the intercept; $(u_i, v_i)$ is the position of sample point $i$; $\beta_0(u_i, v_i)$ is the constant term of sample point $i$; $\beta_k(u_i, v_i)$ denotes the coefficient of the $k$ independent variable of sample point $i$; $x_{ik}$ is the $k$ independent variable of sample point $i$; and $\varepsilon_i$ is the random error.

In this study, the dependent variable was the characteristics of the river network in the Yellow River basin (represented by the multifractal spectral width $\Delta\alpha$), and the independent variable was the topographic feature factor. In the calculation of geographically weighted regression, the accuracy of the model results is largely affected by the kernel and bandwidth. We used a fixed kernel, and the Akaike information criterion (AICc) was used to determine the bandwidth for the analysis, because the fixed kernel approach is more suitable for gridded data [29,43] and the AICc method can solve the problem better and faster [44,45]. In the calculation results, the modified $R^2$ reflects the degree of explanation of the dependent variable by the independent variables, that is, the influence of the topography on the river network characteristics, which was used to test the performance of the model. Local $R^2$ can reflect the extent to which the independent variable explains the dependent variable in each grid. This result can be used to test the local performance of the model results. When the condition number in the calculation result is less than 0 or greater than 30, it indicates that there is local multicollinearity between the independent variables, and the calculation result of the model is unreliable.

## 3. Results

### 3.1. Multifractal Analysis

In this study, we computed the multifractal characteristics of topography and river networks in the Yellow River Basin based on DEM and river network vector data and the multifractal characteristics of topography and river network in 262 grids of 80 km × 80 km and 893 grids of 40 km × 40 km. Before calculating the multifractal characteristics of the study object, it was necessary to determine the scale-free interval of the study object and verify its multifractal nature. The scale-free intervals of the topography and river network of the Yellow River Basin and its divided grid were calculated, as shown in Table 1.

**Table 1.** Scale range for calculating multifractal features in the Yellow River Basin.

| Range | Total Number | Category | The Range of $q$ Value | The Range of Boxes Size $e$ |
|---|---|---|---|---|
| Yellow River Basin | 1 | Topography | $[-30, 30]$, $\Delta q = 1$ | [30 m, 40,000 m], $\Delta e = 500$ |
| | | River Network | $[-21, 21]$, $\Delta q = 1$ | [30 m, 40,000 m], $\Delta e = 500$ |
| 80 km × 80 km grids | 262 | Topography | $[-30, 30]$, $\Delta q = 1$ | [30 m, 20,000 m], $\Delta e = 500$ |
| | | River Network | $[-15, 15]$, $\Delta q = 1$ | [30 m, 20,000 m], $\Delta e = 500$ |
| 40 km × 40 km grids | 893 | Topography | $[-45, 45]$, $\Delta q = 1$ | [30 m, 20,000 m], $\Delta e = 500$ |
| | | River Network | $[-25, 25]$, $\Delta q = 1$ | [30 m, 20,000 m], $\Delta e = 500$ |

The scale range in Table 1 was used to calculate and test whether the study area had multifractal properties. The topography and river network of the Yellow River Basin and its partition grid had a good linear relationship between $\ln M(e, q)$ and $\ln e$, whose Pearson correlation coefficients were all greater than 0.95. In addition, the $\tau(q)$ values of the study objects were convex functions with respect to $q$. The above results indicate that the studied topography and river network are scale-invariant in the selected scale range with prominent multifractal characteristics, which can be further analyzed and studied based on multifractal theory.

The multifractal characteristics of the study object were mainly represented by the generalized fractal dimension ($D_q$) and the multifractal spectrum [$\alpha(q) \sim f(\alpha)$]. They were analyzed according to the spectral width $\Delta \alpha$ of the multifractal spectrum and the difference between the maximum and minimum values $\Delta f$ of the multifractal spectrum. To visualize the multifractal characteristics of the grid delineated in the Yellow River basin, only the $\Delta \alpha$ and $\Delta f$ of the topography and river network in the Yellow River basin at different scales are displayed here. The results are shown in Figure 2, where "T" and "R" represent Topography and Rivers, respectively. For example, "$\Delta \alpha$-T" represents the spectral width of the multifractal spectrum of topography in Figure 2.

The values of $\Delta \alpha$ and $\Delta f$ in multifractals represent different meanings in topography and river networks. (1) When we analyzed topography characteristics, the span of the singularity exponent $\Delta \alpha$ could quantitatively describe the degree of inhomogeneity of the distribution probability of the watershed characteristics. The larger $\Delta \alpha$ is, the worse the uniformity of the distribution of feature information in the basin, the greater the topographic relief (i.e., the greater the variation within the fractals), and the more polarized the probability trend of each subset. The $\Delta f$ was used to calculate the difference between the maximum and minimum distribution probability subset numbers of the basin characteristics information. It indirectly reflects the proportion of the number of peaks and valleys on the basin landform surface. (2) In the analysis of the river network, $\Delta \alpha$ reflected the spatial distribution of morphological variability in the river network. A larger $\Delta \alpha$ indicates greater variation within the fractals and a less uniform spatial distribution of the river network. $\Delta f$ reflects the spatial distribution of the fluvial density.

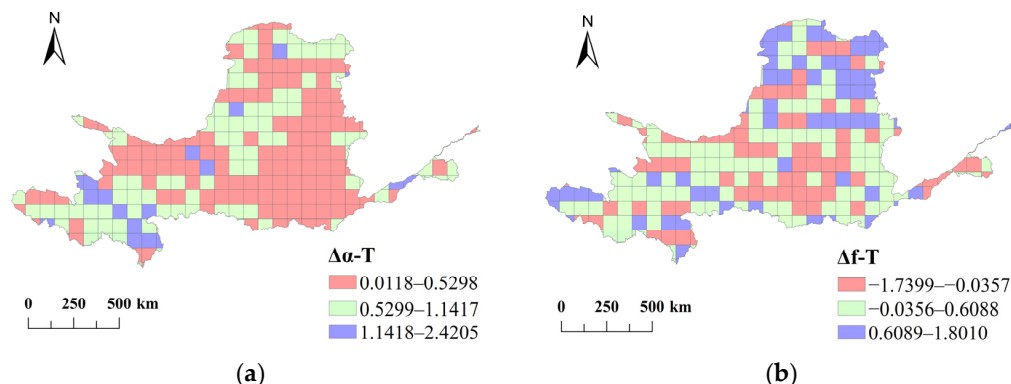

**(a)**                                                              **(b)**

**Figure 2.** *Cont.*

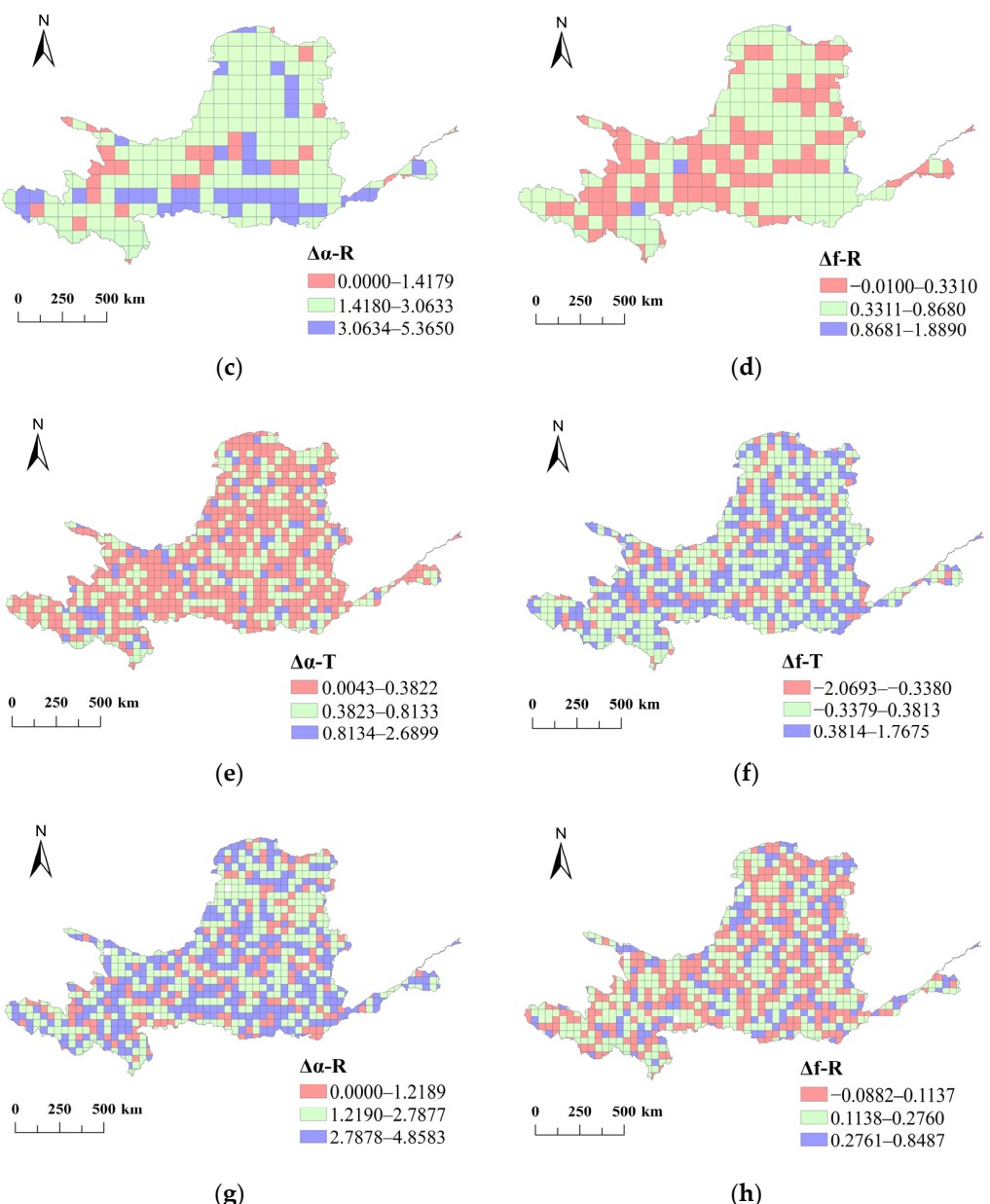

**Figure 2.** Multifractal characteristics of Yellow River Basin. (**a**) $\Delta\alpha$ of an 80 km $\times$ 80 km grid topography; (**b**) $\Delta f$ of an 80 km $\times$ 80 km grid topography; (**c**) $\Delta\alpha$ of an 80 km $\times$ 80 km grid river network; (**d**) $\Delta f$ of an 80 km $\times$ 80 km grid river network; (**e**) $\Delta\alpha$ of a 40 km $\times$ 40 km grid topography; (**f**) $\Delta f$ of a 40 km $\times$ 40 km grid topography; (**g**) $\Delta\alpha$ of a 40 km $\times$ 40 km grid river network; (**h**) $\Delta f$ of a 40 km $\times$ 40 km grid river network.

The different values of $\Delta\alpha$ and $\Delta f$ represent the complexity of the topography and river networks in different regions. The spatial distribution of the multifractal features of the topography and river network in the Yellow River Basin is shown in Figure 2. From the perspective of different scales, the terrain and river network of the Yellow River Basin had strong spatial differentiation, and they showed different degrees of differentiation at different scales. The differentiation decreased with an increase in scale, and the information expressed was more detailed. From the perspective of multifractal characteristics of the terrain and river network, the corresponding river network structure of the region with a more complex terrain distribution was relatively simple, indicating that the terrain and river network showed a negative correlation trend to some extent. Therefore, we assumed that topography had a negative influence on the structure of river networks based on

the above results, and we further study the relationship between topography and river networks in the Yellow River basin in detail and quantitatively in a later chapter.

### 3.2. Geographical Detection Analysis and Influence of Topography on the River Network

Different topographic factors can reflect the information on topographic characteristics from different perspectives. Topographic relief and slope are basic indicators for describing and expressing topographic and geomorphic morphological features [46]. According to the above multifractal characteristics of the Yellow River Basin, it is clear that there was a certain correlation between its topography and river network, and the topography had a certain influence on the structure and morphology of the river network. Therefore, 13 indicators were selected in this study as independent variables (Table 2) from three dimensions, namely multifractal, topographic relief, and slope factor. The multifractal characteristics of the river network ($\Delta\alpha$) were used as dependent variables to represent the structure of the river network. We used a geographic detector to analyze the correlation between topography and the river network and the degree of influence of topography on the fluvial structure.

**Table 2.** Explanatory variables and their description.

| Dimension | Name | Symbol | Unit | Description |
|---|---|---|---|---|
| Multifractal characteristics | The width of the multifractal spectrum | $\Delta\alpha$ | — | It is used to represent the degree of inhomogeneity, irregularity, and complexity of the terrain in the study area. |
| | The difference of the multifractal spectrum | $\Delta f$ | — | It is used to reflect the difference of quantity distribution of the maximum- and minimum-probability subsets of the watershed characteristic information. |
| | Capacity dimension | $D_0$ | — | It is equivalent to the box fractal dimension, reflecting the complexity of the research object. |
| | Information dimension | $D_1$ | — | It adds coverage probability on the basis of the capacity dimension. |
| | Correlation dimension | $D_2$ | — | It is used to reflect the degree of connection between subjects. |
| Topographic relief factors | Average elevation | $H$ | Meter (m) | It represents the average elevation of the region and reflects the overall elevation of the region. It is calculated using a DEM with a resolution of 30 m. |
| | Maximum elevation | $H_{\max}$ | Meter (m) | It represents the maximum elevation of the region and reflects the overall elevation of the region. It is calculated using a DEM with a resolution of 30 m. |
| | Minimum elevation | $H_{\min}$ | Meter (m) | It represents the minimum elevation of the region and reflects the overall elevation of the region. It is calculated using a DEM with a resolution of 30 m. |
| | Topographic relief | $\Delta H$ | Meter (m) | It represents the difference between the maximum and minimum values of regional elevation and reflects the relief of regional terrain. It is calculated using a DEM with resolution of 30 m. |
| | Topographic roughness | $R$ | — | It represents the roughness of the terrain [47]. It is calculated using a DEM with resolution of 30 m. |
| Slope factors | slope | $S$ | Degree (°) | It represents the average slope of the region. It is calculated using a DEM with a resolution of 30 m. |
| | Slope aspect | $SA$ | Degree (°) | It represents the average slope aspect of the region. It is calculated using a DEM with a resolution of 30 m. |
| | Slope length | $SL$ | Meter (m) | It represents the average slope length of the region. It is calculated using a DEM with a resolution of 30 m. |

The topographic relief factor reflects the variation of topography in elevation and the regional morphological characteristics of the terrain from a more macroscopic perspective. The slope factor is a topographic factor that reflects the microscopic form of the slope

and can reflect local changes in the terrain. Multifractals can reflect the complexity of the overall and local morphology of the terrain in more detail from multiple perspectives and contain more topographic feature information, which has always played an important role in topographic analysis [2]. In this study, we used divergence and factor detectors in the geographical detector method to analyze the degree of influence of topography on the structure of the river network from the macroscopic, microscopic, and comprehensive perspectives of topography.

Based on the above-selected variables, the values of the selected variables were calculated in the range of 80 km × 80 km and 40 km × 40 km grids in the Yellow River Basin. Then, the natural interruption method was used to stratify the selected variables. The factor detection of the selected indicators was carried out with geographical detectors, and the detection results are listed in Table 3. At two different grid sizes (80 km × 80 km and 40 km × 40 km), five of the thirteen factors passed the significance test at a 0.01 level; these factors were multifractal spectrum width, multifractal spectrum height difference, average elevation, maximum elevation, and average slope. In addition, at the scale of 40 km, one indicator (minimum elevation) passed the significance test at the 0.05 level, while at the scale of 80 km, the same indicator passed the significance test at the 0.1 level. In summary, six factors passed the significance test at different scales, and the results in Table 3 were obtained by ranking them according to the value of $q$.

**Table 3.** The results of geographical detection.

| Dimension | Explanatory Variable Name | 40 km × 40 km Grids | | | 80 km × 80 km Grids | | |
|---|---|---|---|---|---|---|---|
| | | Significance Level | *q* Value | Rank | Significance Level | *q* Value | Rank |
| Multifractal characteristics | The width of the multifractal spectrum | 0.01 | 0.501 | 1 | 0.01 | 0.407 | 1 |
| | The difference of the multifractal spectrum | 0.01 | 0.265 | 2 | 0.01 | 0.217 | 2 |
| | Capacity dimension | — | 0.263 | — | — | 0.205 | — |
| | Information dimension | — | 0.125 | — | — | 0.135 | — |
| | Correlation dimension | — | 0.002 | — | — | 0.039 | — |
| Topographic relief factors | Average elevation | 0.01 | 0.133 | 4 | 0.01 | 0.108 | 4 |
| | Maximum elevation | 0.01 | 0.124 | 5 | 0.01 | 0.101 | 5 |
| | Minimum elevation | 0.05 | 0.113 | 6 | 0.1 | 0.098 | 6 |
| | Topographic relief | — | 0.050 | — | — | 0.037 | — |
| | Topographic roughness | — | 0.024 | — | — | 0.057 | — |
| Slope factors | slope | 0.01 | 0.245 | 3 | 0.01 | 0.199 | 3 |
| | Slope aspect | — | 0.087 | — | — | 0.018 | — |
| | Slope length | — | 0.046 | — | — | 0.079 | — |

Note: "—" indicates that the significance test was not passed.

The ranking results of the factors that passed the significance test in Table 3 show that the multifractal spectrum width of the terrain was the main factor influencing the structure of the river network, with an explanation rate of more than 50%, which was much higher than other factors. This result indicates that the complexity of the terrain had a greater influence on the structure of the river network. The next-most influential factors were the multifractal spectrum height difference and slope, with an explanation rate of more than 20%, followed by topographic relief, including elevation, elevation maximum, and elevation minimum, with explanatory power also reaching approximately 10%. From different scales of analysis, the $q$ values of geographic detection decreased with an increase in the analysis scale. However, the explanatory power ranking of the influencing factors did not change, and the ranking was multifractal spectrum width > multifractal spectrum height difference > average slope > average elevation > elevation maximum > elevation minimum. The ranking of the explanatory power of the influencing factors did not change

with scale, indicating that the topographic factors affecting the structure of the river network were mainly the six factors that passed the significance test.

### 3.3. Correlation between Topography and River Network Structure

To further study the spatial differences in the magnitude and direction of the topographic factors that passed the significance test on the river network structure in different analysis units, we introduced a local spatial regression analysis of the geographically weighted regression model to investigate the direction and intensity of the six significant topographic factors on the river network and their spatial differences in different analysis units. The results calculated using the geographically weighted regression showed that the corrected $R^2$ of the model was 0.634 at the analysis scale of 80 km, and the range of condition numbers was 16.656–29.959. At an analysis scale of 40 km, the corrected $R^2$ of the model was 0.705, and the range of condition numbers was 17.988–27.999, which indicates that the model passed the multicollinearity test, and the goodness of fit was high. This indicates that the calculated results had a high degree of confidence. The local $R^2$, standard errors of the coefficients, and regression coefficients of each independent variable were calculated, and the results are shown in Figures 3 and 4.

Local $R^2$ indicates the local performance of the model, and the larger the absolute value, the better the effect of the model. Figures 3a and 4a show the distribution of the local $R^2$ of the geographically weighted regression model. It can be seen from the figure that the local $R^2$ of the Yellow River Basin gradually increased from west to east. Meanwhile, Figures 3a and 4a also show that each independent variable had spatial non-stationarity, but the characteristics of its influence degree were different. The standardized residual of the coefficient reflects whether there are local calculation problems in the model. When the standardized residual exceeds 2.5, it indicates problems in the computation of the geographically weighted regression model in this region. It can be seen from Figures 3b and 4b that there were only two regions with more than 2.5 times the standardized residual in the analysis scale of 80 km, but this did not affect the analysis of the overall results, which also indicates the high reliability of the calculation results.

The multifractal spectral width of the terrain represents the singularity and complexity of its spatial distribution. Figure 3c shows that the multifractal spectral width of the terrain was negatively correlated with the dependent variable at the 80 km analysis scale in general and positively correlated in the upper reach of the Yellow River basin. The degree of influence on the fluvial structure increased gradually from west to east with certain regularity. The difference between the maximum and minimum distribution probability subsets of the basin information was indicated by the multifractal spectral height difference. As shown in Figure 3d, the multifractal spectral height difference of topography and the dependent variable showed a positive correlation in general and a negative correlation in the Yellow River downstream basin. Its contribution to the river network structure gradually decreased from west to east in space and became negative in the downstream area of the Yellow River. The negative influence area accounted for approximately 2% of the total area of the Yellow River basin.

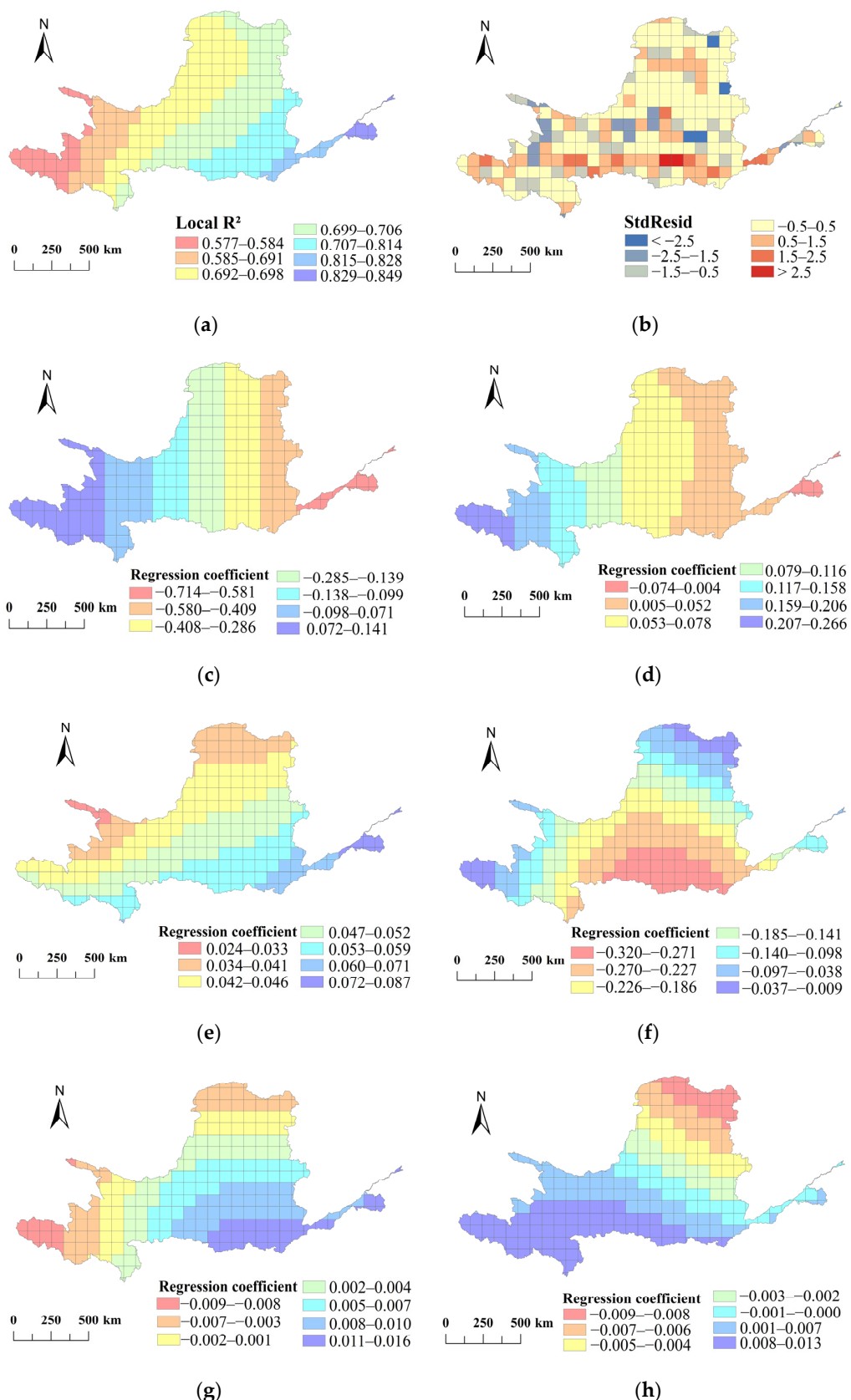

**Figure 3.** Influence degree of topography on river network structure at the 80 km analysis scale. (**a**) Local $R^2$; (**b**) Standardized residual of the coefficient; (**c**) The width of the multifractal spectrum; (**d**) The difference in the multifractal spectrum; (**e**) Slope; (**f**) Average elevation; (**g**) Elevation maximum; (**h**) Elevation minimum.

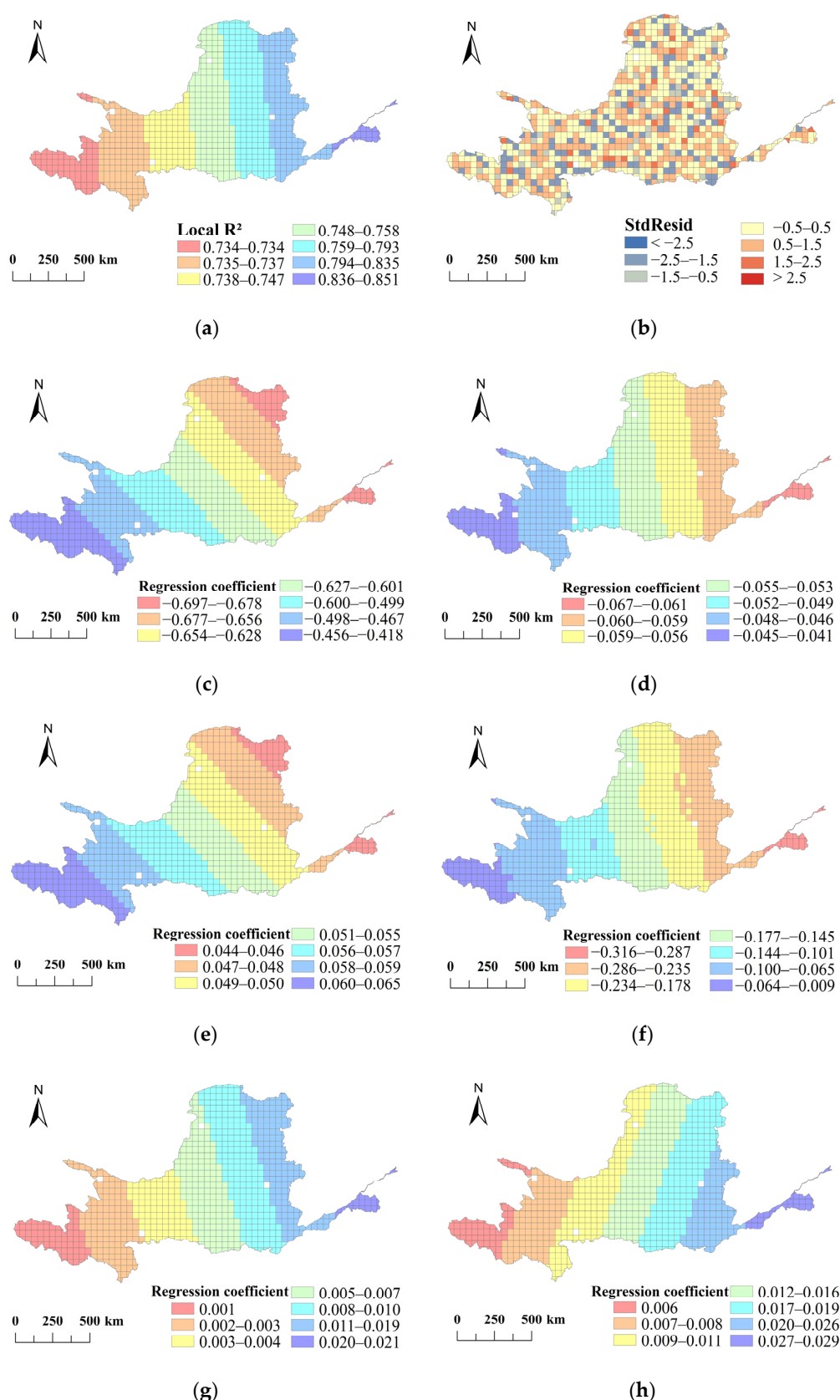

**Figure 4.** Influence degree of topography on river network structure at the 40 km analysis scale. (**a**) Local $R^2$; (**b**) Standardized residual of the coefficient; (**c**) The width of the multifractal spectrum; (**d**) The difference in the multifractal spectrum; (**e**) Slope; (**f**) Average elevation; (**g**) Elevation maximum; (**h**) Elevation minimum.

At an analysis scale of 80 km, the slope had a facilitating effect on the structure of the river network in all analysis units, and the degree of its influence decreased from northwest to southeast. The influence of mean elevation on river network structure was negative, and its influence increased from north to south in a "circular" trend. Meanwhile, the topography of the Yellow River basin gradually decreased from north to south, indicating that the influence of elevation on river network structure was greater in the lower topographic areas. The influence of maximum and minimum elevation on the structure of the river network was generally promoting, but a negative influence was also observed in approximately 30% of the areas in the north and west of the Yellow River basin. The higher-elevation areas indicated that the extreme values of elevation in the higher elevation had a suppressive effect on the structure of the river network. The regression coefficient of the maximum elevation increased gradually from northwest to southeast of the Yellow River basin in a circular pattern, but the degree of its influence decreased first and then increased. Moreover, more areas with a positive influence of the maximum elevation on the structure of the river network than those with a negative influence were observed.

The regression coefficient of the minimum elevation increased from the northeast to the southwest of the Yellow River basin, and the magnitude of its influence also decreased first and then increased. The area where the maximum elevation had a positive influence on the structure of the river network was dominant. The extreme value of elevation had a positive effect on the structure of the river network, which indicates that the complexity of the river network structure was higher in the Yellow River Basin when the topography was more undulating.

Based on the above results, it was found that, at an analysis scale of 80 km, the magnitude and spatial distribution of the degree of influence of different topographic factors on the river network structure were different. Among all the topographic factors affecting the river network structure, the largest regression coefficient (effect) was the width of the multifractal spectrum. Other factors included average elevation, multifractal spectrum height difference, slope, and maximum and minimum elevations. The degree of influence of different topographic factors on the river network differed in spatial distribution, with obvious spatial differences.

Figure 4 shows the degree of influence and spatial variation of different topographic factors on the river network structure at a 40 km analysis scale. It can be declared from Figure 4c that multifractal spectrum width was negatively correlated with river network structure in all analysis units, indicating that topographic complexity had a negative effect on river network structure at this analysis scale, and its effect increased gradually from southwest to northeast, showing significant spatial differences. The influence of the multifractal spectrum height difference on the river network structure was small and negative; its influence degree increased gradually from west to east, and the spatial difference was small, indicating that the number and distribution of the probability subset of terrain characteristic information had little influence on the river network. At this scale, the slope had a positive effect on the river network structure in all units, and its effect was similar to the difference in the multifractal spectrum height. It increased gradually from northeast to southwest, indicating that the slope had a greater influence on river networks in high-terrain areas, such as the Qinghai-Tibet Plateau and the Loess Plateau. The regression coefficient fluctuation of the average elevation was significant, and the spatial heterogeneity of the effect was strong. The effect was negative in all analysis units, and it was most prominent in the eastern part of the Yellow River Basin. Through matching analysis with the elevation distribution in the Yellow River basin, it was found that the lower the terrain, the greater the impact of the average elevation on the river network structure. Although the regression coefficient and fluctuation range of the maximum and minimum elevation values were much smaller than those of the other parameters, the spatial difference was obvious, and their influence on the river network was positive, indicating that topographic relief at this scale promoted the complexity of the river network

structure. The extreme value of elevation varied in different spatial ranges, but the effect size of both gradually increased from west to east.

The influence of different topographic factors on the river network structure and their spatial variation at the 40 km analysis scale showed that, among all kinds of topographic factors, the most influential on the structure of the river network was multifractal spectral width, followed by the mean elevation, and the others are slope, multifractal spectrum height difference, elevation minimum, and elevation maximum, in descending order. The effects of various topographic factors on the river network structure and their spatial distribution were different, with more obvious spatial heterogeneity and regularity, and those effects varied from the west to east in space.

We analyzed the influence of various topographic factors on the structure of the river network in the Yellow River basin at different scales, which showed that there was a high correlation between topography and river network structure, indicating that topography had a greater influence on the development of the river network. The effect direction and magnitude of different topographic factors on the river network were different, and the scale effect of some topographic parameters was obvious. The width of the multifractal spectrum had a negative influence on the river network at different scales. The fluctuation range of the regression coefficient increased gradually with an increase in the analysis scale, whereas the value of the regression coefficient decreased gradually. The multifractal spectral height difference had a negative influence on the river network structure at an analysis scale of 40 km and a positive influence at an analysis scale of 80 km. In all analysis units at different scales, the slope had a positive effect, indicating that the greater the slope, the greater the influence on the river network and the higher the complexity. Additionally, the scale effect of the slope regression coefficient was not significant. The average elevation had a negative effect at different scales, and the scale effect was not obvious. The maximum and minimum elevations mainly had positive effects at different scales, and the scale effect was not significant. Compared with the other topographic parameters, the regression coefficients of the maximum and minimum elevations were smaller, indicating that the effect was weak.

The corrected $R^2$ of the geographically weighted regression model at different scales were 0.634 and 0.705, respectively, which indicates a high correlation between topographic factors and the river network. This also indicates that the river network structure was significantly influenced by topography. Under different scales of analysis, the influence of topographic parameters on the structure of the river network was in the following order: topographic multifractal spectrum width > mean elevation > multifractal spectrum height difference > slope > elevation minimum > elevation maximum. This indicates that the complexity of topography and mean elevation had a greater influence on the formation and development of river networks, while the influence of regional elevation extremes on river networks was relatively weak.

### 3.4. Cluster Analysis of Geographically Weighted Regression Coefficients

The regression coefficients calculated using the geographically weighted regression model for the Yellow River Basin at different analysis scales were second-order clustered. The analysis units of the Yellow River Basin were divided into three categories. At an analysis scale of 80 km, the first category contained 155 analysis units, mainly in the Loess Plateau; the second category contained 32 analysis units, mainly in the lower reaches of the Yellow River basin; and the third category contained 75 analysis units, mainly in the upper reaches of the Yellow River region (mainly in the plateau terrain, including the Qinghai-Tibet Plateau). At the analysis scale of 40 km, the classification results were basically the same as those at the scale of 80 km, with the first category containing 529 analysis units, the second category containing 115 analysis units, and the third category containing 249 analysis units. The clustering results for the Yellow River Basin at different scales are shown in Figure 5. The results show that the clustering results were basically the same, although the analysis scales were different. The classification results were consistent with

different geomorphological units, and the three categories were dominated by different geomorphological units. The first category was dominated by the Loess Plateau and Inner Mongolia Plateau; the second category was dominated by the lower reaches of the Yellow River basin, including the Huang-Huaihai Plain; and the third category was dominated by the Tibetan Plateau. The high agreement of the clustering results with the different geomorphological units also indicates that the formation and development of river network structures in different regions were inextricably related to topographic factors.

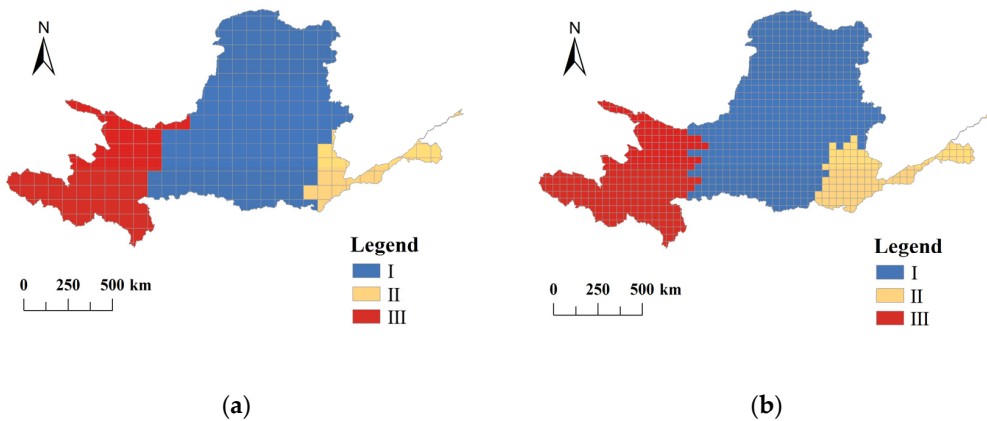

(**a**)　　　　　　　　　　　　　　　　　(**b**)

**Figure 5.** Clustering results of the Yellow River Basin. (**a**) Clustering results at an 80 km scale; (**b**) Clustering results at a 40 km scale.

The results of the second-order cluster analysis are shown in Figures 6 and 7, which indicate that the larger the regression coefficient of a certain type of region for a certain topographic factor, the greater the influence of the topographic factor on the regional river network structure. The clustering results also indicate that the key factors affecting the structure of each regional river network type were different.

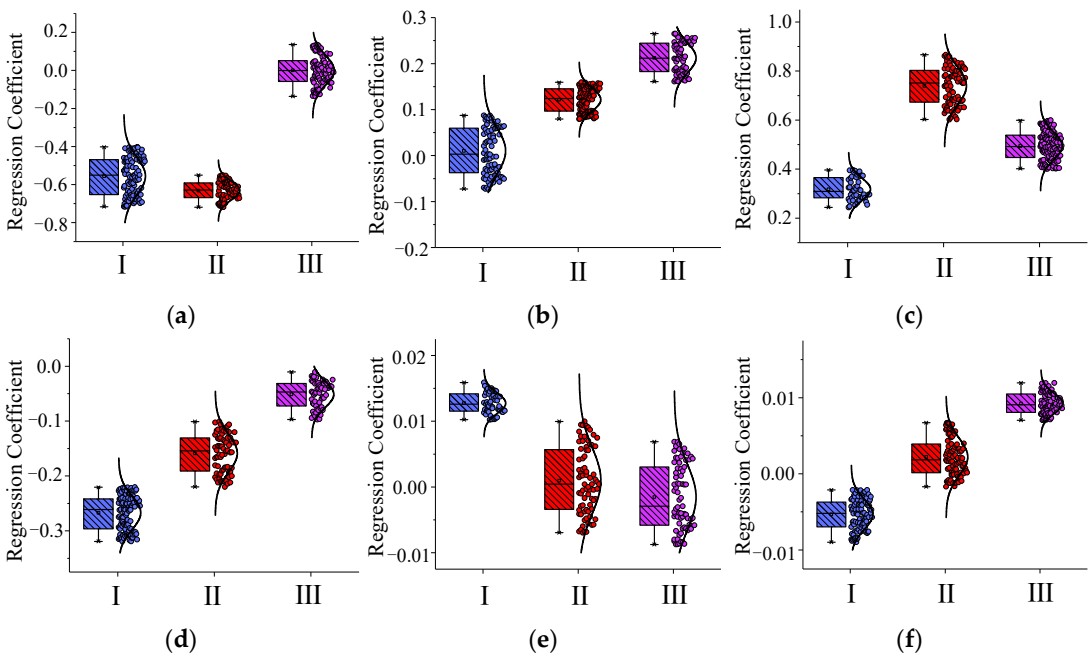

**Figure 6.** Second-order clustering results of the regression coefficients of the geographically weighted regression model at the 80 km analysis scale. (**a**) The width of the multifractal spectrum; (**b**) The difference of the multifractal spectrum; (**c**) Slope; (**d**) Average elevation; (**e**) Elevation maximum; (**f**) Elevation minimum.

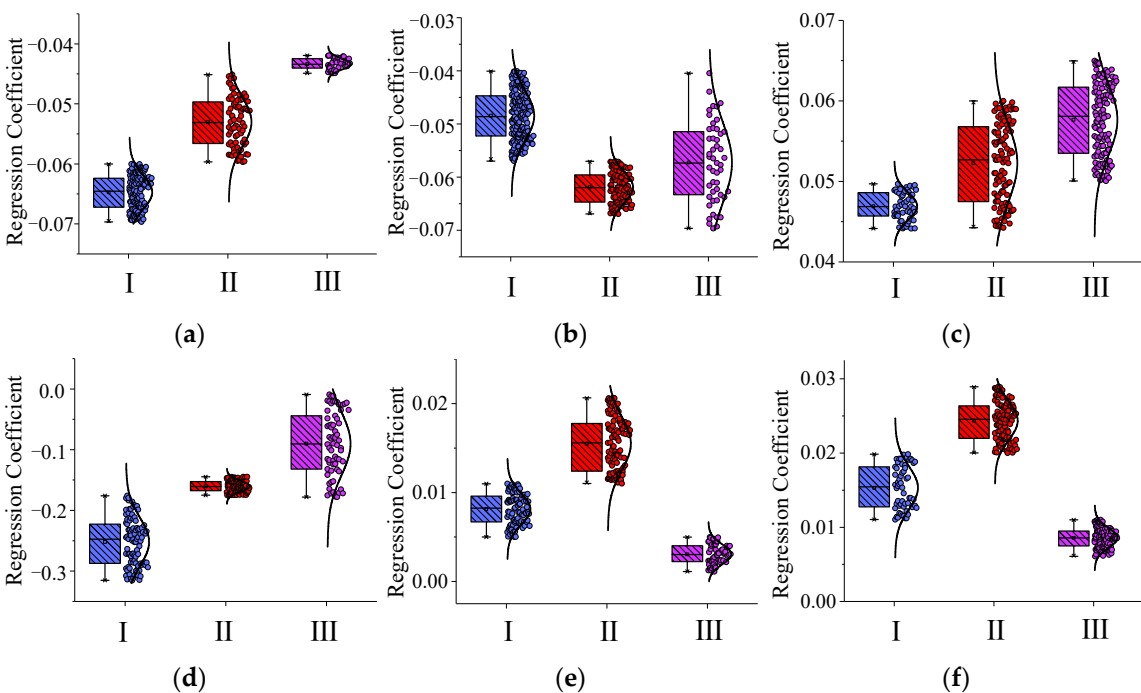

**Figure 7.** Second-order clustering results of the regression coefficients of the geographically weighted regression model at the 40 km analysis scale. (**a**) The width of the multifractal spectrum; (**b**) The difference of the multifractal spectrum; (**c**) Slope; (**d**) Average elevation; (**e**) Elevation maximum; (**f**) Elevation minimum.

The main influencing factors of the different regional river networks were obtained by second-order cluster analysis of the Yellow River Basin. In Figures 6 and 7, it can be seen that, at the scale of 80 km, the river network structure in the first category was influenced by the topographic multifractal spectrum width, average elevation, and elevation maximum; the river network structure in the second category was most influenced by the slope, followed by the multifractal spectrum width; and the third category was most sensitive to the multifractal spectrum difference and elevation minimum. At the scale of 40 km, the river network structure in the first category was mainly influenced by the multifractal spectrum width and average elevation; the river network structure in the second category was most sensitive to the multifractal spectrum difference and elevation minimum; and the river network structure in the third category was most affected by the slope.

The clustering analysis of the Yellow River basin at different scales shows that the influencing factors of the river network structure in the first category were not affected by the scale of analysis, that is, the topographic multifractal spectral width and average elevation. The influencing factors of the river network in the second and third categories varied with the scale of analysis, and the change was more obvious in the third category. Different categories of areas contain different geomorphic units, and this result also explains why the structure of the river network in different geomorphic units was mainly influenced by different topographic factors.

## 4. Conclusions and Discussion

In this study, we calculated and analyzed the spatial variation of multifractal characteristics of the terrain and river network in the Yellow River basin and explored the relationship between topography and river network structure under different analysis scales using a geographic detector and geographically weighted regression models. The main conclusions are as follows:

(1) The river network and terrain of the Yellow River Basin had obvious multifractal properties, and the multifractal characteristics of both had strong fractality in space at dif-

ferent analysis scales. The fractality of their multifractal characteristics gradually decreased with an increase in the analysis scale. The multifractal characteristics of the topography and river network were negatively correlated; that is, the topographic distribution was more complex in the region, and the corresponding river network structure was simpler.

(2) The geographical detection results of the river network structure in the Yellow River Basin show that the six topographic factors that passed the significance test were the main topographic factors affecting the river network structure, and the order of the explanatory power of the influencing factors did not change with the change in scale. The order of the explanatory power of the influencing factors at different analysis scales was as follows: multifractal spectrum width > multifractal spectrum height difference > average slope > average elevation > elevation maximum > elevation minimum.

(3) There was a high correlation between the river network structure and topography in the Yellow River basin. The impact intensity and magnitude of different topographic factors on the river network structure were different. The width of the multifractal spectrum had a negative influence on the river network structure, which also indicates that topographic complexity had a restraining effect on the river network structure complexity.

(4) The second-order clustering of regression coefficients from the results of the geographically weighted regression model of the Yellow River basin revealed that the basin was divided into three types of areas, and the key influencing factors of the river network structure were different in different types of areas. The clustering results were highly consistent with different geomorphological units, and this result also indicates that the river network structure was closely related to geomorphological types.

As basic physical geographic elements, the structure and characteristic information of river networks and topography in basins were spatially complex and variable. Multifractals provide a reliable method and theoretical support for the quantitative representation of their characteristics. Furthermore, there was a close correlation between the river network and topography. We analyzed the correlation between river networks and topography from the perspective of multifractals and the influence of different types of topographic factors on the river network structure at different analysis scales. The results are of great significance for the scientific and quantitative representation of river network characteristics and the development of topography and geomorphology.

**Author Contributions:** Conceptualization, Zilong Qin and Jinxin Wang; methodology, Zilong Qin; software, Zilong Qin; resources, Zilong Qin; data curation, Jinxin Wang; writing—original draft preparation, Zilong Qin; writing—review and editing, Zilong Qin, Jinxin Wang; funding acquisition, Jinxin Wang. All authors have read and agreed to the published version of the manuscript.

**Funding:** This research was funded by the Key Scientific and Technological Project of Henan Province, China (grant number 212102210377).

**Institutional Review Board Statement:** Not applicable.

**Informed Consent Statement:** Not applicable.

**Data Availability Statement:** The data presented in this study are available on request from the corresponding author.

**Acknowledgments:** We would like to thank all editors and reviewers for their profound comments, which helped us to improve the quality of this paper.

**Conflicts of Interest:** The authors declare no conflict of interest.

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
