# Peer review of "Multifractal Correlation between Terrain and River Network Structure in the Yellow River Basin, China"

_ijgi, doi:10.3390/ijgi11100519_

Round 1
Reviewer 1 Report
This manuscript selected the Yellow River Basin in China as the study area, and calculated topographic factors of multiple dimensions. The influence of different topographic factors on the river network structure at different scales and their correlation from a multifractal perspective based on geographical detectors and a geographically weighted regression model were determined. The finding makes some sense, but some details still need to be improved. Specific modification suggestions are as follows:
1) In abstract, it does not seem to allow me to visually see some of the distinctive conclusions. In particular, there are phrases that don't make much sense, such as the last sentence.
2) The introduction aims to critically analyze the existing literature to emphasize the rare or difficult content of the current research that this research solves. However, the introduction of this article is mostly aimed at the local background of the case study, rather than focusing on the theory. In addition, the introduction should not only state the research objectives, but also the author's originality and novelty (which is related to the lack of innovation of the study).
3) The authors should be aware that IJGI is a well-known international journal. Articles published in international journals should focus on universality, not a report to the Chinese authorities. Therefore, some universal meanings need to be further reinforced.
Thus, I suggested “reconsider after major revision”
Author Response
Dear Reviewer,
Thank you for your comments concerning our manuscript entitled “Multifractal Correlation between Terrain and River Network Structure in the Yellow River Basin, China”. We appreciate the time and effort and are grateful for the insightful comments. Those comments are valuable and very helpful for revising and improving our paper. We have studied comments carefully and have made the correction.
The specific corrections can be found in the attachment.
Once again, thank you very much for your warm work.
Best regards.

Reviewer 2 Report
Review of the manuscript (article) titled Multifractal Correlation between Terrain and River Network 2 Structure in the Yellow River Basin, China
In this article the spatial variation of multifractal characteristics of the terrain and river network in the Yellow River basin and explored the relationship between topography and river network structure under different analysis scales using a geographic detector and geographically weighted regression models were calculated and analyzed. The results are of great significance for the scientific and quantitative representation of river network characteristics and the development of topography and geomorphology. Therefore, the article is a valuable contribution to the explanation the processes underlying the development of topographic landforms and river networks. The article is well written.
Below are some suggestions for improving your article.
All references listed in References were cited in the text.
In line 635 (names of all authors are missing) should be written Mukherjee, S.; Mukherjee, S.; Garg, R.D.; Bhardwaj, A.; Raju, P.L.N. instead of Mukherjee, S.; Mukherjee, S.; Garg…, R.
In line 697 (volume and page range are missing) should be written Developments in Soil Science, 2009, 33, 3-30. instead of DEVELOPMENTS IN SOIL SCIENCE 2009.
Most of the cited papers are available online, so it would be good to list DOI.
In Figure 1, I propose to reverse the display of elevation, i.e. green color for low, and brown color for high.
In line 391 should be written Figure 3(c) instead of Figure 3€.
In Figures 4 and 7, the text (description) under each panel should be deleted, it should be as it is in Figures 3 and 6.
In Figure 4(c) – 4(h) should be written Regression coefficient instead of 回归系数.

Author Response
Dear Reviewer,
Thank you for your comments concerning our manuscript entitled “Multifractal Correlation between Terrain and River Network Structure in the Yellow River Basin, China”. Those comments are helpful for revising and improving our paper. We have studied comments carefully and made the correction.
The specific corrections are as follows:
The missing information in those references has been revised, and the DOI of references has been listed.
The display of elevation in Figure 1 has been reversed.
The incorrect writing has been corrected in this paper.
The text under each panel has been deleted in Figure 4 and 7.
Once again, thank you very much for your warm work.
Best regards.
Reviewer 3 Report
The paper is very vell structured.
Author Response
Thank you for your review concerning our manuscript. We appreciate the time and effort that you dedicated to providing feedback on our manuscript.
Round 2
Reviewer 1 Report
This manuscript has been revised and greatly improved. I have no objection to this manuscript now. Therefore, I suggest that it can be accepted.